# Is Self-Reported Physical Fitness Useful for Estimating Fitness Levels in Children and Adolescents? A Reliability and Validity Study

**DOI:** 10.3390/medicina55060286

**Published:** 2019-06-18

**Authors:** Augusto César Ferreira De Moraes, Regina Célia Vilanova-Campelo, Francisco Leonardo Torres-Leal, Heráclito Barbosa Carvalho

**Affiliations:** 1Department of Epidemiology; School of Public Health, University de São Paulo, 01246-904 São Paulo, Brazil; augusto.moraes@fm.usp.br; 2YCARE (Youth/Child cArdiovascular Risk and Environmental) Research Group, Faculdade de Medicina, Universidade de São Paulo, 01246-903 São Paulo, Brazil; regina.vilanova-campelo@usp.br (R.C.V.-C.); torresleal@ufpi.edu.br (F.L.T.-L.); 3DOMEN (Metabolic Diseases, Exercise and Nutrition) Research Group, Department of Biophysical and Physiology, Federal University of Piaui, 64049-550 Teresina, Brazil

**Keywords:** physical fitness, questionnaire, reliability, validity, children, adolescents, self-reported

## Abstract

*Background and objective*: The assessment of physical fitness has become a necessary issue in epidemiological studies, since a reduction in fitness is directly associated with early mortality. Therefore, the development of simple, accurate, and inexpensive methods is necessary to measure physical fitness. This study aimed to determine the reliability and validity of the criteria and constructs of the International Fitness Scale (IFIS), Portuguese version, in Brazilian pediatric populations. *Methods*: A total of 190 children aged 3–10 years and 110 adolescents aged 11–17 years were enrolled in an observational study of reliability and validity. For reliability, the participants completed a questionnaire twice (with an interval of 15 days). To test the criterion validity, we analyzed the agreement between the questionnaire and physical tests (20-m shuttle run test, handgrip strength, standing long jump tests, 4 × 10-m shuttle run test, and back-saver sit and reach test), and the construct validity was estimated by agreement between the questionnaire and high blood pressure. The reliability was analyzed by kappa coefficients. The agreement between the testing and retesting of the questionnaire was evaluated by kappa coefficients. We applied a 2 × 2 table to estimate the specificity, sensitivity, and accuracy of the questionnaire. *Results*: The mean age of the children was 6.7 years (*n* = 190), and for the adolescents it was 14.6 years (*n* = 110). The questionnaire reliability showed an almost perfect score (κ ≥ 0.93 in children and κ ≥ 0.88 in adolescents). The questionnaire showed moderate criterion validity (κ ≥ 0.40 in children and adolescents) as well as moderate construct validity (κ ≥ 0.40) in the components of general conditioning, cardiorespiratory capacity, muscular strength, and speed/agility in children and in the components of cardiorespiratory capacity, muscle strength, and speed/agility in adolescents. The questionnaire was a sensitive method for measuring physical fitness. *Conclusions*: The Portuguese version of the IFIS is a reliable and valid method for measuring physical fitness in pediatric populations.

## 1. Introduction

Physical fitness integrates the measures of most body functions, such as skeletomuscular and cardiorespiratory functions, involved in the performance of daily physical activity and/or physical exercise [1], which is the reason why physical fitness is considered an important health marker [2].

Physical fitness is defined as a state of well-being, which refers to the ability to perform daily tasks, sports, or occupations without undue fatigue [3]. Furthermore, it is recognized as a set of attributes that are either health- or skill-related [4]. The regular practice of physical exercise contributes to the improvement of physical fitness [5]. Studies have indicated that physical inactivity [6] and reduced physical fitness are associated with an increasing number of health problems, mainly obesity, cardiovascular diseases, musculoskeletal health problems, and mental health issues [7,8].

In addition, longitudinal cohort data have indicated that low levels of physical fitness in late adolescence are associated with an increased risk of all-cause mortality in adulthood [9]. On the other hand, high levels of physical fitness in childhood and adolescence are related to more favorable health outcomes [3,10]. These studies demonstrated that physical fitness screening is necessary in health-related educational follow-ups and in epidemiological studies [11].

Fitness tests are usually assessed in laboratories or fields: However, in the school context, the estimated time to perform a physical fitness assessment in 20 children/adolescents is approximately 2 h and 30 min, i.e., three 55-minute physical education classes [10]. Therefore, this type of evaluation remains difficult to perform in the school environment and/or in epidemiological studies due to limited time, space requirements, and material availability [12]. As an alternative, the use of a questionnaire can be considered in the measurement of physical fitness [12].

Ortega et al. proposed a subjective (self-reported) method called the International Fitness Scale (IFIS) to measure physical fitness levels in epidemiological surveys when this evaluation cannot be performed due to time and budget constraints [12]. This scale is available in nine different languages and evaluates general physical, cardiorespiratory, and muscular fitness as well as agility and flexibility [12]. The reliability and validity of the questionnaire in children, adolescents, adults, and other populations have been studied [13,14,15], but before accepting it for global use, it is necessary to validate it in different populations and age groups [12].

Studies have been conducted in older children [15]: However, it is unknown whether the IFIS is a reliable and valid instrument in 3–8-year-olds. To validate a Portuguese version of the IFIS would be useful given that it is the sixth most spoken language (native speakers) in the world. Beyond language translation, cultural adaptation of this kind of tool for different countries needs to be studied. To the best of our knowledge, this questionnaire has been previously used in South American samples [12,14,15], without its validity having been studied in Brazilians, so this study is of special interest in interpreting previous findings in these countries and advising of its usefulness in future studies. Examining physical fitness from an early age may contribute to the development of strategies to prevent the reduction of physical fitness throughout life [16]. Furthermore, no reliability and validation study of this questionnaire with a field test and blood pressure (BP) has been identified in the pediatric population of South America, specifically in Brazilians. Hence, the purpose of the present study was to determine the reliability and criterion validity of the International Fitness Scale (IFIS), Portuguese version (IFIS-LP) in Brazilian pediatric populations and the IFIS-LP construct validity to predict high blood pressure in children and adolescents.

## 2. Materials and Methods

We conducted an observational study of reliability and validity. This study used data from a random sample of children aged 3–10 years and adolescents aged 11–17 years [17] who were enrolled in preschool, primary school, and up to the third year of high school at both public and private schools in the city of Teresina, Piauí, Brazil, who participated in the South American Youth/Child Cardiovascular and Environmental (SAYCARE) study [18]. The exclusion criteria were an inability to perform physical tests (pregnancy, use of medications for blood pressure, heart conditions, or joint pain) and/or respond to questionnaires, and/or refusing to sign the informed consent. This consent was necessary for the parents or guardians and for the adolescents or children. In addition, the headmasters of the selected schools also gave their consent to collaborate with the study. Three schools were chosen due to convenience for data collection. Their headmasters were contacted and received a formal invitation with detailed information about the study. For the schools that agreed to participate, an information letter and verbal explanation were provided for the potential participants and their parents or guardians.

To assess the reliability of the questionnaire [12] (self-reported physical fitness), measurements were taken from the same subjects on two occasions with a 15-day interval, following international protocols [19]. For children, the parents or guardians filled out the questionnaire [12,20], whereas the adolescents answered the questions by themselves. The criterion validity of the questionnaire was evaluated by comparing it to the physical test (described below). To test the construct validity, we analyzed the questionnaire responses with high blood pressure prevalence. The health indicator chosen was high blood pressure [21], as low levels of physical fitness are associated with the development of cardiovascular diseases [22]. Additionally, systemic arterial hypertension is an independent risk factor for cardiovascular diseases [21]. Maternal education was classified according to years of school, as reported by the questionnaire. The options for maternal education according to years of school were <4 years, 4–8 years, 9–12 years, or >12 years [23].

### 2.1. Measures

#### 2.1.1. Subjective Measurement Methods (Self-Reported Physical Fitness)

Self-reported physical fitness was evaluated through the IFIS [12], the International Fitness Scale, in the Portuguese language version (IFIS-LP). The IFIS was originally written in English and then translated and culturally adapted (reverse translation) [24]. After translation permission, the IFIS-LP underwent cross-cultural adaptation for the production of the Portuguese version of the questionnaire, according to the methodology proposed by Herdman [25]. The IFIS contains five items related to general physical fitness, cardiorespiratory fitness, muscular fitness, speed/agility, and flexibility levels, with answers scored on a five-point Likert scale (very poor, poor, average, good, and very good). The questions address an individual’s self-perception of physical fitness evaluated against that of his/her friends. In cases of doubt, the research team was able to clarify some of the components of physical fitness. For agreement analysis, the results of the Likert scale responses for the five IFIS-LP questions were dichotomized into good/great (average, good, and very good) and low/very low (poor and very poor) for all bands, according to recommendations in the literature [26].

#### 2.1.2. Objective Measurement Methods

##### Anthropometry

The anthropometric variables of weight and height were analyzed according to the reference manual of anthropometric standardization of the World Health Organization (WHO) [27]. Body mass index (BMI) was calculated as an individual’s body weight in kilograms divided by the square of his/her height in meters [27]. We used an ultra Slim W801 digital scale (Crivitta Diagnostica Ltd, São Paulo, Brazil) and Wall stadiometer (Cardiomed, Paraná, Brazil). Anthropometric variables were measured once and then later repeated. An additional measurement was performed in cases of an error of 5% between the first and the second measurements. The evaluations were conducted in a private room at the school. All measures were taken in underwear or with as few clothes as possible and without shoes.

##### Blood Pressure (BP) Measurements

We measured systolic (SBP) and diastolic (DBP) blood pressure with a calibrated mercury column sphygmomanometer coupled to an Omron device (Omron Health Care, Japan), model HEM-7200, with an appropriate cuff for each arm size. We used three different cuffs according to the following anthropometric measurements of the arm: 12 to 21 cm (small), 22 to 32 cm (medium), and 33 to 42 cm (large) [28]. The measurements were performed on the right arm of the participants because of coarctation of the aorta, and the arm was supported at the level of the heart: All were placed in a quiet room. The children and adolescents sat, their backs resting on a chair, a relaxed arm resting on a rigid surface, and uncrossed feet resting on the floor. After 5 min of rest, the measurement was initiated. BP was determined by averaging two measurements taken within 5 min of each other, with the subject resting for at least 5 min before the first measurement [29]. High blood pressure (HBP) was defined as SBP and DBP above the 95th percentile for sex, age, and height following the American Academy of Pediatrics protocol [30].

##### Physical Fitness Tests

To ensure objectivity of the measurements of physical fitness, a detailed manual of operations on the field-based fitness tests was read by every researcher involved in the fieldwork before data collection started. In addition, all researchers performed one week of training in order to standardize and harmonize measurement of the physical fitness tests. Physical tests were performed after BP was measured. All physical tests were performed individually, in the following order. Muscular fitness (MF) was assessed using two tests [31]: (1) The handgrip test (maximum handgrip strength assessment) using a hand dynamometer with an adjustable grip (Jamar® PC5030J1, Fit Systems Inc., Calgary, Canada). The participant stays in a standard bipedal position with the arms in complete extension holding the dynamometer without touching any part of the body with it. Scores were calculated as the average of the right and left handgrip strength used in the analysis [26]. Two trials were allowed for each hand, and the average score was recorded in kilograms (kg). (2) The standing broad jump test (lower limb explosive strength assessment). The participant jumps as far as possible off the stand, trying to land with both feet together and maintain equilibrium once landed (they were not allowed to put their hands on the floor). The score was obtained by measuring the distance between the last heel mark and the take-off line. The best of two attempts was recorded in centimeters [32]. We computed the MF variable as the average of the two MF tests. Flexibility was measured by the sit-and-reach test for range of movement (in cm) [32]. Speed and agility were assessed with the 4 × 10-m shuttle run test, a test where the participant runs as fast as possible from the starting line to the other line and returns to the starting line (10 m apart), crossing each line with both feet every time. This was performed twice and covered a distance of 40 m (4 m × 10 m). Every time the child crossed any of the lines, he/she picked up (the first time) or exchanged a sponge that had been previously placed behind the lines. The time taken to complete the test was recorded to the nearest tenth of a second [32]. Cardiorespiratory fitness (CRF) was assessed by the 20-m shuttle run test [33]. Participants were required to run between two lines 20 m apart, while keeping pace with audio signals emitted from a prerecorded compact disc. The group of children was evaluated using a modified version of the original test [33], reducing the initial velocity to 6.5 km/h. For the adolescents, the initial speed was 8.5 km/h, which was increased by 0.5 km/h (1 min = 1 stage) [26]. Participants were encouraged to keep running as long as possible throughout the course of the test, and it was completed when the participants failed to reach the end lines concurrent with the audio signals on two consecutive occasions. The last half-stage completed was recorded as an indicator of his or her CRF. Overall fitness was computed as the average of all four physical fitness components studied [12].

The results of the children’s physical tests were used to classify this group according to good/great physical fitness (≥90th percentile for age and sex in this sample) and low/poor physical fitness (below the 90th percentile). Based on a reference for European children, a Likert-type scale to classify children´s performance was as follows: (very poor (*X* < percentile 10), poor (percentile 10 ≥ *X* < percentile 25), good (percentile 75 ≤ percentile 90), and very good (≥ percentile 95)) [34]. For the adolescent results, we used age- and sex-adjusted cutoffs [2,10,34].

### 2.2. Data Collection

Data collection occurred during five school visits:

(1) An explanation of the project and the delivery of informed consent to be signed by parents/guardians or the adolescents themselves;

(2) The delivery of the questionnaire (parents or adolescent self-completion) for an evaluation of self-reported physical fitness, blood pressure measurements, and anthropometric measurements;

(3) The collection of the completed questionnaire (self-reported physical fitness) and an objective evaluation of physical fitness (field test);

(4) The second delivery of the questionnaire in order to assess reliability; and

(5) The collection of the completed questionnaire (second application) and a second objective evaluation of physical fitness (field test (we considered the second evaluation for analysis)).

### 2.3. Statistical Analysis

Sample size calculations were performed to verify the reliability and validity of the self-reported physical fitness questionnaire in the study population, for which we used values from the cardiorespiratory fitness physical test [33]. For the reliability analysis, the sample size was calculated by using a nomogram [35], and the parameters used were an alpha (α) of 0.05 (type I error), a beta (β) or power (type II error) of 0.80, and a kappa coefficient (κ) of 0.70. For the validity analysis, the parameters were as follows: an α of 0.05 (type I error) (two-tailed), a β or power (type II error) of 0.80, and a kappa coefficient of 0.80. From these parameters, the necessary sample size estimated was 135 participants for the reliability analysis and 119 for the validity analysis. Considering the possible loss of participants, a 10% greater sample size was recruited for these analyses (*n* = 149 for reliability and *n* = 131 for validity). The descriptive analysis included calculating the mean, the percentage, and their respective 95% confidence intervals (95% CI). The normality of the sample was observed through the Shapiro–Wilk test. 

The test–retest reliability of the self-reported physical fitness measurements was calculated for categorical variables by using the kappa statistic. Kappa coefficients (κ ≥ 0.40) [36] were considered acceptable. To test the validity of the criterion, we analyzed the agreement between the questionnaire and the physical test, and the validity of the construct was calculated by the concordance of the questionnaire between high blood pressure and the kappa coefficient, where moderate (or higher) values of the kappa concordance coefficient ≥0.40 [36] were considered acceptable. It was assumed that good/great physical fitness in the questionnaire would be in agreement with good/great physical fitness in the physical test and with normal pressure (by measuring blood pressure). For a complementary validity analysis and to understand if the questionnaire discriminated between the children and adolescents according to the field test, we applied a Kruskal–Wallis ANOVA to compare the fitness field test results across the questionnaire categories.

We applied a 2 × 2 contingency table, and the sensitivity and specificity of the questionnaire were considered: Sensitivity, which was the proportion of participants with good/great physical fitness according to both instruments ((self-reported physical fitness and physical fitness tests)–(proportion of participants with normal blood pressure and good/great physical fitness according to both instruments)); specificity ((proportion of participants with very poor physical fitness, poor according to both instruments)–(proportion of participants with high blood pressure and very poor physical fitness, poor according to both instruments)); prevalence (participants with good/great physical fitness according to the questionnaire); and accuracy (proportion of participants correctly diagnosed by both instruments). Stata software version 14.0 (StataCorp, College Station, Texas) was used for all statistical analyses.

### 2.4. Ethical Considerations

This study was approved by the Research Ethics Committee of the University of São Paulo Medical School, filed under number (58930816.7.0000.0065). Written informed consent was obtained from all of the participants in the study.

## 3. Results

The sample of this study was composed of a pediatric population aged 3 to 17 years. One-hundred and ninety children and 110 adolescents participated in the study, with a total of 300 participants evaluated. Table 1 shows the descriptive variables. The mean age of the children was 6.7, and for adolescents, it was 14.6 years.

Self-reported physical fitness is presented in Table 2. Most parents/guardians of the children rated their children’s physical fitness as acceptable or good physical fitness. Most adolescents reported acceptable physical fitness.

Table 3 shows the test–retest reliability statistics in children and adolescents from Teresina Piauí, Brazil, for the five items that comprise the IFIS, that is, overall fitness and the four main fitness components: CRF, MF, speed and agility, and flexibility. The reliability coefficients were acceptable for all components of physical fitness in the group of children and adolescents.

Table 4 presents the validity of the self-reported physical fitness based on the agreement between the measures of the questionnaire and the physical test (criterion validity). The coefficient of validity showed an acceptable value for all components of physical fitness in both groups of children and adolescents.

The sensitivity and specificity of the self-reported physical fitness are shown in Table 5. The questionnaire presented greater sensitivity and lower specificity in the group of children and adolescents when compared to the field tests.

Table 6 presents the physical fitness field test results according to the physical fitness questionnaire categories in both groups of children and adolescents. Children classified as very good presented higher results in the field tests for all of the fitness components. On the other hand, in adolescents, only for flexibility was the result not significant when we compared the very good to the very poor category.

The construct validity (sensitivity and specificity) of the questionnaire compared to blood pressure is shown in Table 7. The questionnaire presented greater sensitivity and lower specificity in the group of children and adolescents, except in the components of general conditioning and flexibility in the group of children that presented greater specificity. The sensitivity of the questionnaire represented the proportion of individuals who had the outcome “good/great physical fitness”, while the specificity of the questionnaire represented the proportion of individuals who did not have that outcome. These values are presented in more detail in Table 7. The means of performance from the physical fitness tests in children and adolescents are shown in the Appendix A.

## 4. Discussion 

This study was performed to establish the reliability and validity of a physical fitness self-report questionnaire, the IFIS (International Fitness Scale, Portuguese version), in a pediatric population, since the reduction of this aptitude is directly associated with early mortality [37]. The physical fitness questionnaire showed almost perfect reliability coefficients in both age groups. The criterion validity coefficient was acceptable in both the children and adolescents. In children, the questionnaire showed moderate construct validity in the components of general conditioning, cardiorespiratory capacity, muscular strength, and speed/agility. In adolescents, the questionnaire demonstrated moderate construct validity in the components of cardiorespiratory capacity, muscular strength, and speed/agility. The questionnaire was shown to be a sensitive method to measure physical fitness in the pediatric population.

These results indicate that the IFIS questionnaire (Portuguese language version) may be useful for estimating the level of physical fitness in the pediatric population, as it is a powerful tool in epidemiological research since it can be used on a large scale, at low cost, and with easy logistics.

Our results showed that parents/guardians and adolescents understood the IFIS issues. Studies have indicated that younger individuals have greater flexibility [38]. Flexibility tends to decrease with age, which may be related to distinct patterns of routine use of major body joints throughout life [39]. In the group of children, flexibility was classified as one of the physical fitness components with the best performance. These results showed that the parents/guardians of the children understood the questions in the questionnaire and were able to respond to them in a way that was very close to the real one, since they were in agreement with the literature [38,39]. Studies have indicated that the younger the age, the greater the flexibility [38]. Flexibility tends to decline after age 17, partly as a result of a decline in physical activity and normal aging [40]. The same was observed among adolescents, and general physical fitness was indicated as one of the best components in this group. The components of physical fitness are inseparable from fundamental movement skills [41,42]. Rarely, or perhaps never, does the individual perform a movement activity that does not involve some aspect of strength, speed, or flexibility. As adolescents are in a specialized movement phase [43], it is assumed that during physical activity and physical exercise, they realize that their performance is strictly related to the acquisition of all components of physical fitness. This may indicate that parents/guardians and adolescents did not randomly choose between the options in questionnaire responses.

The physical fitness questionnaire showed reliability coefficients of near perfect agreement in the groups of children and adolescents. These results indicate that the questionnaire had an acceptable reliability in measuring physical fitness in the pediatric population. Previous studies performed with children between 9 and 12 years of age, adolescents, and adults have presented lower values of reliability [12,14,15,44]. In our study, we evaluated children 3 years of age and older: In this age group, parents and/or guardians closely follow the growth and development [43] of their children. In this sense, by observing them alone and in a group, parents tend to also have a greater knowledge of the children who study with their children.

Another explanation for the reliability of the results of the questionnaire is that people may have a better subjective perception of their strength than when asked about physical activity. The physical activity questionnaire generally covers different domains [45]. Therefore, many questions are needed to investigate the construct of interest, which can lead to errors, overlapping responses, and reduced attention and motivation [46]. On the other hand, in the IFIS-LP, there are only five questions for the construction of five variables [12]. Due to the nature of the very specific IFIS-LP issues, the answers are less likely to overlap or confuse, thus reducing memory bias and increasing the chance of bias of the desired behavior. Most likely, physical fitness questionnaires [47,48,49,50] have better results than physical activity questionnaires because it is easier to have a subjective perception of the physical valences required to perform physical effort than to report behaviors related to physical activity [51]. The IFIS was designed with the principle that the human brain may be more accurate in classification than in quantification [12], so it is easier to classify people into categories (e.g., normal or overweight) than to estimate their current weight.

We found a moderate validity of the physical fitness questionnaire when compared to the physical test. The findings of the present study indicate that the IFIS-LP questionnaire has an acceptable validity in measuring physical fitness in children and adolescents when compared to physical tests (criterion validity). The performance of the instrument in relation to the physical test can be explained in the literature by the correlation between the evaluated outcomes [32,52]. Recent studies have also indicated an acceptable validity of the physical fitness questionnaire compared to physical tests [12,14,15]. Our results indicate that there is a concordance of the criteria measured in the physical fitness questionnaire when compared to the physical tests.

The IFIS questionnaire, as a physical fitness identifier, was verified according to sensitivity parameters (efficiency in identifying the presence of physical fitness) and specificity (efficiency in identifying the absence of physical fitness). Simpler tests can be used as substitutes for more elaborate but more accurate or precise forms of establishing the presence of an outcome and/or disease [53,54]. However, we must consider that there was a risk of error in the classification of these results, for example, rating one with the presence of physical fitness when in fact it was absent. However, this risk was justified by the safety and convenience of simpler tests, such as the questionnaire used in this study. In addition, childhood daily play routines require adequate physical conditioning to perform physical activities [41]. The involvement of adolescents in physical activities can transform the prognosis of this health indicator [55] because the evolution of physical fitness can be perceived during the activities of daily living [3]. Our results indicate that the questionnaire can be used as a screening test.

The agreement between the measurements of physical fitness and high blood pressure supports the construct validity of this questionnaire. Cardiorespiratory fitness is related to high blood pressure [56]. Our results in this study showed that even with a low prevalence of high blood pressure between children and adolescents (Table 1), the results of sensitivity and accuracy were good, since it is more difficult to have good accuracy with a low prevalence [57]. The questionnaire presented high sensitivity, showing good results in assessing people with good/great physical fitness even with a low prevalence of high blood pressure. Cardiorespiratory fitness is a predictor of high blood pressure [56]. Low levels of speed/agility, balance, muscle strength, and cardiorespiratory fitness are significantly related to the onset of high blood pressure [58].

We can highlight some of the strengths of this study. Among them was a comparison between the IFIS physical fitness questionnaire and objective measures (physical tests and blood pressure) in a pediatric population in a middle-income city, with an established, standardized, and poorly explored methodology. Cross-cultural adaptation was performed in order to avoid semantic problems [59], since when the reverse translation was performed, there was agreement among the researchers involved in this evaluation. The validation of instruments requires time, an adequate sample (*n*), statistical tests, and cultural adaptation procedures [60]. Therefore, this study contributes significantly to the literature in this area of research.

Although the study had strengths, it had limitations that should be discussed. One of the possible limitations of the present study was that the parents who answered the questionnaire in the group of children (3–10 years) may have overestimated the physical fitness of the evaluated ones [12], since the parents self-reported the physical fitness of their children in comparison to the physical fitness of their peers. Another possible limitation was the selection of the sample by convenience and the small prevalence of HBP, which could have been affected by the construct validity. However, since the convenience selection included public and private schools, this type of choice should not have influenced the reliability and validity of the results, since pre-established criteria were used to guarantee presumed socioeconomic representativeness as well as age and sex. The strength of the present study was to evaluate physical fitness through field tests. However, by using the same physical tests used by Ortega et al., these tests can be widely used and tested for validity and reliability [12].

## 5. Conclusions

The Portuguese version of the IFIS is a reliable and valid method for estimating physical fitness in pediatric populations. We recommend the use of the IFIS-LP in measuring physical fitness in pediatric populations due to its practicality, low cost, and easy logistics.

## Figures and Tables

**Table 1 medicina-55-00286-t001:** Basic characteristics of the studied children and adolescents.

	Children (*n* = 190)	Adolescents (*n* = 110)
	Mean or % (SD)	Mean or % (SD)
Age	6.7 (2.1)	14.6 (1.8)
Body weight (kg)	25.8 (11.3)	51.7 (11.4)
Height (cm)	119.1 (15.5)	158.4 (12.0)
BMI (kg/m^2^)	17.5 (3.4)	20.7 (5.6)
Systolic blood pressure (mmHg)	95.5 (9.9)	109.9 (5.3)
Diastolic blood pressure (mmHg)	64.3 (8.4)	65.5 (4.0)
High blood pressure ^a^ (%)	1.9	2.9
Maternal education level (%)		
Incomplete high school	16.0	23.3
High school	21.5	40.0
Technical education	8.3	6.67
University degree	54.2	30.0

SD: standard deviation; BMI: body mass index. High blood pressure was defined as systolic blood pressure or diastolic blood pressure above the 95th percentile for sex, age, and height.

**Table 2 medicina-55-00286-t002:** Distribution of responses by categories of self-reported physical fitness (*) in children and adolescents.

**Components**	**Children (*n* = 190)**				
	**Very good % (*n*)**	**Good % (*n*)**	**Average % (*n*)**	**Poor % (*n*)**	**Very poor % (*n*)**
Overall fitness	15.8 (30)	26.8 (51)	28.4 (54)	27.9 (53)	1.1 (2)
Cardiorespiratory fitness	12.1 (23)	24.7 (47)	31.2 (59)	28.9 (55)	3.1 (6)
Muscular strength	18.9 (36)	20.0 (38)	31.6 (60)	25.8 (49)	3.7 (7)
Speed and agility	18.9 (36)	33.7 (64)	32.6 (62)	14.3 (27)	0.5 (1)
Flexibility	8.4 (16)	22.1 (42)	46.9 (89)	20.0 (38)	2.6 (5)
**Components**	**Adolescents (*n* = 110)**				
	**Very good % (*n*)**	**Good % (*n*)**	**Average % (*n*)**	**Poor % (*n*)**	**Very poor % (*n*)**
Overall fitness	5.5 (6)	24.5 (27)	62.7 (69)	5.5 (6)	1.8 (2)
Cardiorespiratory fitness	9.1 (10)	18.2 (20)	50.9 (56)	15.5 (17)	6.3 (7)
Muscular strength	1.0 (1)	30.9 (34)	58.2 (64)	8.1 (9)	1.8 (2)
Speed and agility	7.3 (8)	30.9 (34)	53.6 (59)	7.3 (8)	0.9 (1)
Flexibility	5.4 (6)	20.9 (23)	50.0 (55)	14.6 (16)	9.1 (10)

* IFIS-LP: International Fitness Scale, Portuguese version.

**Table 3 medicina-55-00286-t003:** Test–retest (one week apart) reliability of the physical fitness questionnaire (*) in the pediatric population from Teresina Piauí, Brazil.

Components	Children (*n* = 190)		Adolescents (*n* = 110)	
	Agreement %	κ	Agreement %	κ
Overall fitness	99.47	**0.99**	96.36	**0.93**
Cardiorespiratory fitness	98.95	**0.98**	99.18	**0.97**
Muscular strength	96.32	**0.95**	96.36	**0.93**
Speed and agility	98.95	**0.98**	92.73	**0.88**
Flexibility	95.26	**0.93**	93.64	**0.90**

* IFIS-LP: International Fitness Scale, Portuguese version, applied at two different times with an interval of 15 days; κ: Moderate (or above) kappa concordance coefficient (κ ≥ 0.40) values are in bold.

**Table 4 medicina-55-00286-t004:** Criterion of validity of the physical fitness questionnaire (*) in the pediatric population.

Components	Children (*n* = 190)		Adolescents (*n* = 110)	
	Agreement %	*K*	Agreement %	*k*
Overall fitness	82.63	**0.60**	82.73	**0.49**
Cardiorespiratory fitness	86.26	**0.65**	81.82	**0.40**
Muscular strength	76.26	**0.50**	83.84	**0.43**
Speed and agility	79.47	**0.50**	86.36	**0.54**
Flexibility	70.00	**0.40**	84.55	**0.53**

* IFIS-LP: International Fitness Scale, Portuguese version, applied at two different times with an interval of 15 days; κ: Moderate (or above) kappa concordance coefficient (κ ≥ 0.40) values are in bold; criterion validity: Estimated by the agreement between the questionnaire and the physical test.

**Table 5 medicina-55-00286-t005:** Sensitivity and specificity of the physical fitness questionnaire (*) compared to field tests in the pediatric population.

**Components**	**Children (*n* = 190)**			
	**Sensitivity (95% CI)**	**Specificity (95% CI)**	**Prevalence (95% CI)** **Good/Great AF**	**Accuracy (95% CI)**
Overall fitness	89.60 (85.26–93.94)	69.23 (62.67–75.79)	75.79 (69.04–82.54)	84.05 (78.03–89.80)
Cardiorespiratory fitness	95.20 (92.16–98.24)	66.15 (59.43–72.88)	67.09 (60.34–73.84)	85.79 (79.04–88.54)
Muscular strength	96.84 (94.36–99.33)	53.68 (46.59–60.77)	70.53 (63.78–77.28)	75.2 3(68.48–77.98)
Speed and agility	94.40 (91.13–97.67)	50.77 (43.66–57.88)	85.26 (78.51–92.01)	79.96 (74.21–83.71)
Flexibility	94.62 (91.42–97.83)	46.39 (39.30–53.48)	77.35 (70.60–84.50)	70.45 (64.01–73.51)
**Components**	**Adolescents (*n* = 110)**			
	**Sensitivity (95% CI)**	**Specificity (95% CI)**	**Prevalence (95% CI)** **Good/Great AF**	**Accuracy (95% CI)**
Overall fitness	95.06 (91.01–99.11)	48.28 (38.94–57.61)	92.73 (85.98–99.48)	83.43 (77.68–86.18)
Cardiorespiratory fitness	91.95 (86.87–97.04)	43.48 (34.21–52.74)	79.09 (71.49–86.69)	82.79 (77.04–87.54)
Muscular strength	90.70 (85.27–96.13)	50.00 (40.66–59.34)	90.00 (83.25–96.75)	82.38 (76.95–86.45)
Speed and agility	95.40 (91.49–99.32)	52.17 (42.84–61.51)	91.82 (85.07–98.57)	75.52 (69.77–78.27)
Flexibility	91.46 (86.24–96.69)	50.00 (40.66–59.34)	76.85 (70.01–83.60)	81.55 (75.41–86.69)

IFIS-LP: International Fitness Scale, Portuguese version; CI: Confidence interval; AF: Physical fitness; sensitivity (proportion of participants with good/great AF in the questionnaire and field test); specificity (proportion of participants with low/poor AF in the questionnaire and field test); accuracy (proportion of participants correctly diagnosed by both methods).

**Table 6 medicina-55-00286-t006:** Physical fitness field test results according to the physical fitness questionnaire categories in children and adolescents.

**Components**	**Children (*n* = 190)**				
	**Very Good**	**Good**	**Average**	**Poor**	**Very Poor**
	**Median (IQR)**	**Median (IQR)**	**Median (IQR)**	**Median (IQR)**	**Median (IQR)**
Cardiorespiratory fitness (mt)	**704 (645–880)**	755 (658.5–884.5)	833 (696–943)	730 (565–857)	**661 (632–1001)**
Muscular strength (kg)	**11.5 (8–13.5)**	10 (7–13.5)	8 (6–12)	6 (4–11.5)	**7 (5.5–9.5)**
Speed and agility (seg)	**13.7 (13.2–15.2)**	14.6 (13.2–17)	16.1 (13.7–19.7)	16.9 (14–19)	**17.6 (15.2–24.9)**
Flexibility (cm)	**28 (24 –31)**	23 (20–29)	24.5 (20–28)	25 (19–29)	**20 (18.5–24)**
**Components**	**Adolescents (*n* = 110)**				
	**Very Good**	**Good**	**Average**	**Poor**	**Very Poor**
	**Median (IQR)**	**Median (IQR)**	**Median (IQR)**	**Median (IQR)**	**Median (IQR)**
Cardiorespiratory fitness (mt)	**665 (614–768)**	625 (574–750)	674.5 (616–725.5)	**600 (555.5–766)**	**605.5 (620–778)**
Muscular strength (kg)	**19 (16.5–29)**	**26.5 (21.5–33)**	23.5 (18–28)	19.8 (17–23.5)	**14 (23–25)**
Speed and agility (seg)	**8 (4–12.9)**	12.6 (4–14.6)	12.3 (4–14.2)	14.1 (12.7–15.6)	**14.6 (14–15)**
Flexibility (cm)	22 (15–30)	20.5 (17.5–27)	22 (16–26)	18 (15–23)	18 (7–22)

SD: Standard deviation; IQR: Interquartile range; * IFIS-LP: International Fitness Scale, Portuguese version. In bold, differences (*p* < 0.05) between the groups.

**Table 7 medicina-55-00286-t007:** Physical fitness questionnaire accuracy, sensitivity, and specificity (construct validity) to predict high blood pressure (HBP) in the pediatric population.

**Fitness Components**	**Children (*n* = 190)**		
	**Sensitivity (95% CI)**	**Specificity (95% CI)**	**Accuracy (95% CI)**
Overall fitness	79.78 (71.66–87.90)	100.00 (100.00–100.00)	76.79 (75.55–80.40)
Cardiorespiratory fitness	93.26 (88.19–98.33)	40.00 (30.10–49.90)	89.00 (86.90–94.40)
Muscular strength	88.76 (82.38–95.15)	80.00 (71.91–88.09)	87.08 (82.90–90.30)
Speed and agility	88.76 (82.38–95.15)	60.00 (50.10–69.90)	86.38 (81.12–90.70)
Flexibility	78.65 (70.37–86.94)	100.00 (100.00–100.00)	74.16 (73.00–80.25)
**Components**	**Adolescents (*n* = 110)**		
	**Sensitivity (95% CI)**	**Specificity (95% CI)**	**Accuracy (95% CI)**
Overall fitness	81.48 (68.02–94.94)	60.00 (43.03–76.97)	78.88 (65.39–88.57)
Cardiorespiratory fitness	85.19 (72.88–97.49)	60.00 (43.03–76.97)	81.87 (68.38–91.53)
Muscular strength	77.78 (63.37–92.18)	80.00 (66.14–93.86)	77.96 (62.07–83.59)
Speed and agility	88.89 (78.00–99.78)	60.00 (43.03–76.97)	84.37 (71.84–94.05)
Flexibility	92.59 (83.52–100.00)	40.00 (23.03–56.97)	83.66 (73.29–93.13)

* IFIS-LP: International Fitness Scale, Portuguese version; high blood pressure was defined as systolic blood pressure or diastolic blood pressure above the 95th percentile for sex, age, and height; sensitivity (proportion of participants with good/great physical fitness according to the questionnaire and HBP); specificity (proportion of participants with low/poor physical fitness according to the questionnaire and HBP); accuracy (proportion of participants correctly diagnosed through both methods).

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
