# Peer review of "Is Self-Reported Physical Fitness Useful for Estimating Fitness Levels in Children and Adolescents? A Reliability and Validity Study"

_medicina, 2019, doi:10.3390/medicina55060286_

Round 1
Reviewer 1 Report
Dear authors,
Thank you for your revised manuscript.
However, I was disappointed to see that you did not really adress any of the major comments raised and did not even acknowledge these limitations in the revised manuscript.
For instance:
i) you mention several times that low fitness is a risk factor for later impaired health, yet you sought to examine the association of excellent fitness (not poor fitness) with blood pressure. This is inconsistent with the literature.
ii) The classification of good/optimal fitness does still not make any sense. In your response you mention that fitness > 75 percentile is considered as good according to Miguel-Etayo et al. Why then use 90 percentile for good/optimal?
iii) Still, the conclusion is way to optimistic for the results.
“The Portuguese version of IFIS is a reliable and valid method for estimating
physical fitness in the pediatric population. We recommend the use of IFIS-LP to measure
physical fitness in pediatric population, due to its practicality, low cost and easy logistic”.
I think it is needed to give a more cautious interpretation given all the limitations of the study
iv) Limitations regarding the low sample size and the large variation in age are not adress and have not been acknowledged satisfactory.
v) Also, I think it is problematic that only 4 children and 3 adolescents were classified with high blood pressure. This makes decreases overall power, but according to me also increases the risk of outliers have an enormous effect on the results. This has not been address either.
Overall, this has been a rather disappointing read and authors have (according to me) failed to provide strong argument on why this papers is not in need of any revision (apart from very superficial grammatical edits).
Author Response
Response to Reviewer 1 Comments
Altogether, the manuscript provides some interesting results. However, there are some major limitations, flaws, and unclarities authors need to address.
Response:
Major comments
Point 1: My main concern about this paper is that some of the handlings of data seems a bit inappropriate and that it appears that the analysis has been conducted in a way that limits the comparison to previous IFIS studies. Finally, although analyses seem reasonable at first glance, they seem more appropriate for large samples. Authors should perhaps consider to revise the data analysis to be suitable for smaller samples such as this one.
Response 1: Thank you for your comments. In the previous versions we followed the recommend statistical approach for our hypothesis, as the review can see in this reference: Zaki R, Bulgiba A, Ismail R, Ismail NA. Statistical methods used to test for agreement of medical instruments measuring continuous variables in method comparison studies: a systematic review. PLoS One. 2012; 7: e37908. And recently our research group published a systematic review paper with metanalysis (Nascimento-Ferreira MV, De Moraes ACF, Toazza Oliveira PV, Rendo-Urteaga T, Gracia-Marco L, Forjaz CLM, Moreno LA, Carvalho HB. Assessment of physical activity intensity and duration in the paediatric population: evidence to support an a priori hypothesis and sample size in the agreement between subjective and objective methods. Obes Rev. 2018 Jun;19(6):810-824.) that we tested the minimum sample for validity and reliability tests, and we founded sample sizes between 50 and 99 subjects showed measurements that were stable in both magnitude and interpretation.
However, for better understanding in the new version we present a figure with the results from both applied measurements methods in each category.
Point 2: For instance, why did the authors choose 90th percentile to define low/poor optimal fitness. This is appearing quite unorthodox. Studies suggest that it is important not to be unfit, rather than being extremely fit (which can represent the 90th percentile). Indeed, the definition of being unfit in the important studies by Blair et al. was the lowest quintile (i.e. 20th percentile). Authors need to strongly motivate this cut-off or revised it to be more aligned to previous research.
Response 2: Thank you for your comments. We understanding that cut-off point can be unorthodox, however in this age group do not have a stablished cutoff point and we followed the reference values to classify physical fitness in the group of children, according Etayo et al. International Journal of Obesity. 2014; 38, S57–S66 can be considered the percentile. We adopted good/optimal physical fitness (≥ 90th percentile for age and sex in this sample) because we are evaluating healthy and younger children than those considered by the authors.
Point 3: Also, the data analysis seems to be quite different to previous studies that have validated IFIS (for instance, Ortega FB et al. Int J Epidemiol. 2011 Jun;40(3):701-11 and Ramírez-Vélez R et al. PeerJ. 2017;5:e3351). It would be very helpful that authors aligned their data analysis to be more comparable to previous studies and if not very strongly motivated their reasons to be make their results easy comparable. For instance, why did the author dichotomize the fitness variable (lines 113) into good/great and low/bad in the reliability analysis? When only using these two categories, it is quite obvious that the reliability will be extremely high. Ramirez et al and Ortega et al. did use all categories and reported exact agreement and a weighted kappa. I think this information would be of greater interest for the reader. Also, agreement between measured and self-reported fitness will very high with only two categories (especially with a 10/90 % distribution).
Response 3: Thank you for your comments. As previously stated, the dichotomization was required few participants reported having a very poor or very good physical fitness level. Sánchez-López et al. Scand J Med Sci Sports 2015; 25: 543–55 and Ramírez-Vélez et al. PeerJ; 2017; 5:e3351, they used only four categories for the same reason of our study. However, to able our study comparable with previous study as reviewer suggest, we present in the new version the field tests results in each questionnaire category.
Point 4: Following the aforementioned point, authors may want to clarify why the created a high blood pressure variable rather than using the continuous data similar to Ortega et al. This of special relevance considering the very few cases with of high blood pressure. Indeed, data from Table indicates that there are few children (n≈4) and adolescents (n≈3) adolescents with high blood pressure. Hence, given the few cases the relevance of this analysis may be questioned.
Response 4: Thank you for your comments. We agree with the reviewer; in our sample size the prevalence of high blood pressure is lower. However intend to offer other researchers the possibility of considering IFIS as a predictor for high blood pressure in pediatrics populations, because the prevalence of this health indicator rise up in the last ten years (De Moraes AC, et al Medicine 2013) and because of this we think the results is pretty important as we can see in the sensitivity and specificity indicators.
Point 5: There a relatively few children (n=190) in the study considering the large age span (3-11 years). If ages are evenly distributed, it would mean around 10 boys and girls per year in this age span. The 90th percentile (to classify high fitness) would then mean the child with the best fitness in age class. It seems like this approach are rather sensitive to deviations between ages and schools. Authors needs provide a more detailed explanation on how they classified fitness and also comments on possible limitation with this approach or consider other approaches utilizing more of the data (for instance, creating more cases or using the full-range of data)
Response 5: Thank you for your comments. We test post-hoc, and our sample is large enough to test the validity and reliability, as we described in the Methods section.
We think have a misunderstanding here, we are so sorry for that. The range of the age is 3 to 10 years old, and our hypothesis is not to test the validity and reliability in each age, because this does not make sense, because is important to show that our methods are valid for the children population. And another important point we answered in point 1 sample sizes between 50 and 99 subjects showed measurements that were stable in both magnitude and interpretation and our sample have twice this.
Point 10: Table 1. Confidence intervals for descriptive data is not very helpful. Please provide means and standard deviations or proportions.
Response 10: Thank you for your suggestion. We include the means and standard deviations or proportions.
Table 1. Basic characteristics of the studied children and adolescents. | ||
Children (n = 190) | Adolescents (n = 110) | |
Mean or % (SD) | Mean or % (SD) | |
Age | 6.7 (2.1) | 14.6 (1.8) |
Body weight (kg) | 25.8 (11.3) | 51.7 (11.4) |
Height (cm) | 119.1 (15.5) | 158.4 (12.0) |
BMI (kg/m2) | 17.5 (3.4) | 20.7 (5.6) |
Systolic blood pressure (mmHg) | 95.5 (9.9) | 109.9 (5.3) |
Diastolic blood pressure (mmHg) | 64.3 (8.4) | 65.5 (4.0) |
High blood pressurea | 1.9 (0.8) | 2.9 (0.9) |
Maternal education level | ||
Incomplete high school | 16.0 | 23.3 |
High school | 21.5 | 40.0 |
Technical education | 8.3 | 6.67 |
University degree | 54.2 | 30.0 |
SD, standard derivation. BMI: body mass index; aHigh blood pressure was defined as systolic blood pressure or diastolic blood pressure above the 95th percentile for sex, age and height. |
Sincerely,
Prof. Dr. Heraclito Carvalho
Associate Professor
Departamento de Medicina Preventiva
Faculdade de Medicina, Universidade de Sao Paulo, Sao Paulo, SP, Brazil
YCARE (Youth/Child cArdiovascular Risk and Environmental) Research Group

Reviewer 2 Report
Thank you for your considered responses to the original review.
Author Response
Reviewer 2 Comments
Thank you for your considered responses to the original review.
Response 1: Thank you for your comments.
This manuscript is a resubmission of an earlier submission. The following is a list of the peer review reports and author responses from that submission.
Round 1
Reviewer 1 Report
Vilanova-Campelo et al. investigated the reliability and validity of the Portuguese version of the International Fitness Scale (IFIS) for self-report of physical fitness in Brazilian children and adolescents.
Altogether, the manuscript provides some interesting results. However, there are some major limitations, flaws and unclarities authors need to address.
Major comments
My main concern about this paper is that some of the handling of data seems a bit inappropriate and that it appears that the analysis has been conducted in a way that limits comparison to previous IFIS studies. Finally, although analyses seem reasonable at a first glance, they seem more appropriate for large samples. Authors should perhaps consider to revise the data analysis to be suitable for smaller samples such as this one.
For instance, why did the authors choose 90th percentile to define low/poor optimal fitness. This is appearing quite unorthodox. Studies suggest that it is important not to be unfit, rather than being extremely fit (which can represent the 90th percentile). Indeed, the definition of being unfit in the important studies by Blair et al. was the lowest quintile (i.e. 20th percentile). Authors need to strongly motivate this cut-off or revised it to be more aligned to previous research.
Also, the data analysis seems to be quite different to previous studies that have validated IFIS (for instance, Ortega FB et al. Int J Epidemiol. 2011 Jun;40(3):701-11 and Ramírez-Vélez R et al. PeerJ. 2017;5:e3351). It would be very helpful that authors aligned their data analysis to be more comparable to previous studies and if not very strongly motivated their reasons to be make their results easy comparable. For instance, why did the author dichotomize the fitness variable (lines 113) into good/great and low/bad in the reliability analysis? When only using these two categories, it is quite obvious that the reliability will be extremely high. Ramirez et al and Ortega et al. did use all categories and reported exact agreement and a weighted kappa. I think this information would be of greater interest for the reader. Also, agreement between measured and self-reported fitness will very high with only two categories (especially with a 10/90 % distribution).
Following the aforementioned point, authors may want to clarify why the created a high blood pressure variable rather than using the continuous data similar to Ortega et al. This of special relevance considering the very few cases with of high blood pressure. Indeed, data from Table indicates that there are few children (n≈4) and adolescents (n≈3) adolescents with high blood pressure. Hence, given the few cases the relevance of this analysis may be questioned.
There a relatively few children (n=190) in the study considering the large age span (3-11 years). If ages are evenly distributed, it would mean around 10 boys and girls per year in this age span. The 90th percentile (to classify high fitness) would then mean the child with the best fitness in age class. It seems like this approach are rather sensitive to deviations between ages and schools. Authors needs provide a more detailed explanation on how they classified fitness and also comments on possible limitation with this approach or consider other approaches utilizing more of the data (for instance, creating more cases or using the full-range of data)
It will not be not very easy for the average reader to navigate between the tables (especially Tables 3-6) which are very comparable. Authors can try to work a bit more with the heading of the Tables and also on the explanations in the text. Also, why is supplementary table 1 presented a supplementary table? I would rather focus on the agreement and kappa statistics rather that sensitivy/specificity given the few cases in the study.
Line 34-35 and lines 375-379. Conclusions are way to optimistic to reflect moderate criterion and construct validity. Authors should provide a more cautious conclusion although I agree that self-reported fitness can be useful.
Lines 79. Author state that “this study used data from a random sample”. However, authors write on line 87-88 that “three schools were chosen by convenience for data collection”
Lines 209-211. Was consent collected from parents as well? A 3-year-old cannot provide written consent. Please clarify
Table 1. Confidence intervals for descriptive data is not very helpful. Please provide means and standard deviations or proportions.
Minor comments
Line 26. “The reliability was analyzed by correlation coefficients”. I might be incorrect, but I think it says kappa coefficients in the rest of the manuscript. Please double-check.
Line 16, 44, 274. As I understand, results from these studies indicate that lower levels of physical fitness is related to negative health outcomes, but not reduction (i.e. indicate multiple measures over time).
Line 57. The highlight of European researchers read a bit odd. Sounds strange. Revise to for instance, Ortega et al. have proposed….
Line 113. Bad fitness and very bad fitness ready quite strange. What about poor, very poor or low very low?
Line 160. First, I think overall fitness sounds be better than general fitness. Secondly, please clarify how overall fitness was calculated.
Lines 166-175. It seems like you measured fitness twice. Please clarify which data were used in the analysis.
Author Response
Response to Reviewer 1 Comments
Altogether, the manuscript provides some interesting results. However, there are some major limitations, flaws and unclarities authors need to address.
Response:
Major comments
Point 1: My main concern about this paper is that some of the handling of data seems a bit inappropriate and that it appears that the analysis has been conducted in a way that limits comparison to previous IFIS studies. Finally, although analyses seem reasonable at a first glance, they seem more appropriate for large samples. Authors should perhaps consider to revise the data analysis to be suitable for smaller samples such as this one.
Response 1: We appreciate the reviewer’s concerns. This study integrates the methodology from the “South American Youth/Child Cardiovascular and Environmental” (SAYCARE) Study (17). This Study aimed to develop methods to collect reliable, comparable, and validated data about cardiovascular health biomarkers, lifestyles, environmental, social, and familial factors in children. Therefore, we adopted the same methodology as the SAYCARE study, broadening the original use of the IFIS for the age group of children (3 to 10 years).
Point 2: For instance, why did the authors choose 90th percentile to define low/poor optimal fitness. This is appearing quite unorthodox. Studies suggest that it is important not to be unfit, rather than being extremely fit (which can represent the 90th percentile). Indeed, the definition of being unfit in the important studies by Blair et al. was the lowest quintile (i.e. 20th percentile). Authors need to strongly motivate this cut-off or revised it to be more aligned to previous research.
Response 2: Thank you for your comments. We used the reference values for children, which were established by harmonized procedures of fitness in the literature. Based on reference for European children (Miguel-Etayo et al. International Journal of Obesity. 2014; 38, S57–S66, a Likert type scale to classify children´s performance X, as follows: [very poor (X <P10); poor (P10⩽X<P25); good (P75⩽P90); and very good (⩾P95)], we adopted good/optimal physical fitness (≥ 90th percentile for age and sex in this sample).
Point 3: Also, the data analysis seems to be quite different to previous studies that have validated IFIS (for instance, Ortega FB et al. Int J Epidemiol. 2011 Jun;40(3):701-11 and Ramírez-Vélez R et al. PeerJ. 2017;5:e3351). It would be very helpful that authors aligned their data analysis to be more comparable to previous studies and if not very strongly motivated their reasons to be make their results easy comparable. For instance, why did the author dichotomize the fitness variable (lines 113) into good/great and low/bad in the reliability analysis? When only using these two categories, it is quite obvious that the reliability will be extremely high. Ramirez et al and Ortega et al. did use all categories and reported exact agreement and a weighted kappa. I think this information would be of greater interest for the reader. Also, agreement between measured and self-reported fitness will very high with only two categories (especially with a 10/90 % distribution).
Response 3: Thank you for your comments. The dichotomization of the data was necessary because few participants reported having a very poor or very good physical fitness level. This method is similar with the methodology applied by Sánchez-López et al. Scand J Med Sci Sports 2015; 25: 543–55 and Ramírez-Vélez et al. PeerJ; 2017; 5:e3351, because they used only four categories. The categorization of the data was also necessary to do the sensitivity and specificity analysis.
Point 4: Following the aforementioned point, authors may want to clarify why the created a high blood pressure variable rather than using the continuous data similar to Ortega et al. This of special relevance considering the very few cases with of high blood pressure. Indeed, data from Table indicates that there are few children (n≈4) and adolescents (n≈3) adolescents with high blood pressure. Hence, given the few cases the relevance of this analysis may be questioned.
Response 4: Thank you for your comments. This manuscript integrates a larger study: the SAYCARE Study (17), therefore we prefer to use the same methodology used in the study of Araújo-Moura el al. Obesity; 2018; 26: S41-46. The use of this methodology also enabled the sensitivity and specificity analysis. We described the prevalence of high blood pressure in the table 1 [Children= 1.9% (n=4) and adolescents= 2.9% (n=3].
Point 5: There a relatively few children (n=190) in the study considering the large age span (3-11 years). If ages are evenly distributed, it would mean around 10 boys and girls per year in this age span. The 90th percentile (to classify high fitness) would then mean the child with the best fitness in age class. It seems like this approach are rather sensitive to deviations between ages and schools. Authors needs provide a more detailed explanation on how they classified fitness and also comments on possible limitation with this approach or consider other approaches utilizing more of the data (for instance, creating more cases or using the full-range of data)
Response 5: Thank you for your comments. We apologize, but we considered children (3 to 10 years of age) in our study. (Page 5, lines 200-209) Sample size calculations were performed to verify the reliability and validity of the self-reported physical fitness questionnaire in the studied population, using values of the cardiorespiratory fitness physical test (33). For the reliability analysis, the sample size was calculated using ae nomogram (35), the parameters used were: α of 0.05 (type I error), a beta or power (type II error) of 0.80 and kappa coefficient of 0.70. For the validity analysis, the parameters were as follows: α of 0.05 (type I error) two-tailed, a β or power (type II error) of 0.80 and kappa coefficient of 0.80. From these parameters, the necessary sample size estimated was 135 participants for reliability analysis and 119 for validity analysis. Considering possible losses of participants, an extra of 10% of the calculated sample size was recruited for these analyses (n = 149 for reliability and n = 131 for validity). Recently, Nascimento et al. published an article describing that the sample size le (50 to 99 individuals) to subjectively assess the and duration and intensity of physical activity in the pediatric population (adopting objective methods as reference) provide stable estimates of agreement between methods. As previously explained, there is a scarcity of reference values for children, using harmonized measures of fitness in the literature. Based on reference for European children (Miguel-Etayo et al. International Journal of Obesity. 2014; 38, S57–S66 [very poor (X <P10); poor (P10⩽X<P25); good (P75⩽P90); and very good (⩾P95)], we adopted good/optimal physical fitness (≥ 90th percentile for age and sex in this sample).
Point 6: It will not be not very easy for the average reader to navigate between the tables (especially Tables 3-6) which are very comparable. Authors can try to work a bit more with the heading of the Tables and also on the explanations in the text. Also, why is supplementary table 1 presented a supplementary table? I would rather focus on the agreement and kappa statistics rather that sensitivy/specificity given the few cases in the study.
Response 6: Thank you for your suggestions. We modified the header of table 3 (page 6, lines 247-249) “Table 3. Test-retest (1 week apart) reliability of the physical fitness questionnaire* in the pediatric population from Teresina Piauí, Brazil” and the supplementary table was included in the text (table 7), page 8, lines 274-278.
Point 7: Line 34-35 and lines 375-379. Conclusions are way to optimistic to reflect moderate criterion and construct validity. Authors should provide a more cautious conclusion although I agree that self-reported fitness can be useful.
Response 7: Thank you for your comments. We modified the conclusions: Page 10, line 383-385 “The Portuguese version of IFIS is a reliable and valid method for estimating physical fitness in the pediatric population. We recommend the use of IFIS-LP to measure physical fitness in pediatric population, due to its practicality, low cost and easy logistic”.
Point 8: Lines 79. Author state that “this study used data from a random sample”. However, authors write on line 87-88 that “three schools were chosen by convenience for data collection”
Response 8: Thank you for your comments. The schools were chosen by convenience, however the classes and the children were chosen randomly.
Point 9: Lines 209-211. Was consent collected from parents as well? A 3-year-old cannot provide written consent. Please clarify
Response 9: Thank you for this question. As described by Carvalho HB et al. (2018), for all subjects [(3-10 years) and adolescents (11-17 years)] we collected the consent from parents. (Page 2, lines 90-91) Moreover, children and adolescents had to provide their consent to participate in the study as well.
Point 10: Table 1. Confidence intervals for descriptive data is not very helpful. Please provide means and standard deviations or proportions.
Response 10: Thank you for your suggestion. However, the confidence intervals are informative in epidemiological studies because they can show the precision of the measurement. This is an important reason why we would like to maintain the confidence intervals in table 1.
Minor comments
Point 01: Line 26. “The reliability was analyzed by correlation coefficients”. I might be incorrect, but I think it says kappa coefficients in the rest of the manuscript. Please double-check.
Response 01: Thank you for your correction. We apologize, and It is now corrected in the main document. Line 26 “The reliability was analyzed by Kappa coefficients”.
Point 02: Line 16, 44, 274. As I understand, results from these studies indicate that lower levels of physical fitness is related to negative health outcomes, but not reduction (i.e. indicate multiple measures over time).
Response 02: Thank you for your comments. (Page 2, lines 76-77). Yes. In children, a high degree of tracking would suggest early measurement and intervention as a strategy to assure healthy levels of physical fitness and in later years(16).
Point 03: Line 57. The highlight of European researchers read a bit odd. Sounds strange. Revise to for instance, Ortega et al. have proposed….
Response 03: Thank you for your correction. We apologize, and It is now corrected in the main document. (Page 2, Line 61) “Ortega et al. have proposed a subjective (self-reported) method...”
Point 04: Line 113. Bad fitness and very bad fitness ready quite strange. What about poor, very poor or low very low?
Response 04: We decided to change the terms as suggested (Page 3, line 123) “...and low/very low (poor and very poor) for all the...”.Thank you.
Point 05: Line 160. First, I think overall fitness sounds be better than general fitness. Secondly, please clarify how overall fitness was calculated.
Response 05: First: We replaced the term “general” by “overall” in the new version of the manuscript (Page 4, Line 180 and table 2, 3, 4, 6,7). Secondly: We considered the “overall physical" variable as the average of all four physical fitness components (Cardiorespiratory fitness, Muscular strength, Speed and agility, Flexibility). (Page 4, lines 180-181) “Overall fitness was computed as the average of all four physical fitness components studied.”
Point 06: Lines 166-175. It seems like you measured fitness twice. Please clarify which data were used in the analysis.
Response 06: Thank you for this question. (Page 4, line 198) “we considered the second evaluation for analysis”.
Sincerely,
Regina Célia Vilanova-Campelo, on behalf of coauthors
PhD Candidate at Faculdade de Medicina, Universidade de Sao Paulo

Reviewer 2 Report
Brief Summary:
The authors have sought to determine the reliability and validity of a Portugese-translated IFIS questionnaire. The used commonly employed physical fitness measures in children aged from 3 years and adolescents as the reference standard. In addition, they appear to have used BP as well. Overall, the questionnaire appears reliable and moderately valid for use in this population.
Broad Comments:
Thank you for allowing me to your review your work and congratulations on completing this study. I have specific comments outlined below that I feel will strengthen your manuscript overall but here are my broad comments:
What is the rationale for including 3-year old children in the study? Is there some evidence to suggest 3-year old's current physical fitness is correlated to future health outcomes? You have pointed out this association in adolescence but their is little rationale provided as to why the children (especially the very young ones) were included.
How were the participants blinded to their individual results? Did the parents communicate with the children between the questionnaire completion (first time) and the physical fitness test? Similarly, were the results of the physical fitness test (the first time) made available to participants (or their parents) between the two testing dates? How did you limit the issue of biasing the second tests?
Similar to the comment above, were the physical fitness tests randomized or done in a set order each time?
Did the students complete the shuttle test in groups? If so, how many were in a group, were they randomized into groups and where they were along the line to prevent running with a friend etc.?
It is unclear how you determined physical fitness scores. You listed it as the average of the four physical fitness tests - is this the average of the percentile they were in for each test or the average of the raw score on each test? Why use the average of four - is this a fair representation of physical fitness? As there is no reference provided so I assume this is unique to your study, I think you need to justify your rationale for this a little more.
I am unsure as to why you referenced to blood pressure as not everyone with good cardiorespiratory fitness has good blood pressure. There might be a good reason for it, but it wasn't clear to me why you did this. I know you have referenced it in the discussion but the rationale should be provided prior to the discussion in my opinion so the reader understands why you have used this method.
The tests you used for physical fitness are not gold standard (e.g. 20m shuttle run is not as accurate as a caridopulmonary exercise test for the determination of cardiorespiratory fitness). Some thought should be given to highlighting this as a limitation. Despite this, you still report good reliability and moderate validity.
Specific Comments
Abstract
Nothing to comment on. Well written.
Introduction:
General comments:
Consider defining physical fitness better - specifically highlighting the parameters you measured and their association with health. I think this would strengthen the rationale.
Consider adding more justification for choosing the age ranges you did, especially the young children. I don't think enough evidence has been provided to justify assessing physical fitness in 3-year olds. Just a line or two would be useful.
p2, line 48 - What ages do you define childhood and adolescence?
line 49 - This paragraph seems to be unfinished. What is the point you are trying to make?
line 51 - what does a physical fitness assessment involve?
line 62 - these references, 12-14, would seem to be good comparisons for your results. Have you considered comparing your results against them in the discussion (especially the children and adolescent ones)?
Line 66 - why 3-7 year olds? I thought you examined 3-10year olds? Maybe a typo?
line 70 - needs referencing (other South American samples) or is this a general use of it in the public?
Materials and Methods
pg3, line 118, What scales and stadiometer were used? Brand name, company etc.?
line 131 - I assume all BP was determined by averaging the two measurements, not just high BP? Consider rewording this section (you could remove the word high in this sentence and just define the acronym HBP in the following sentence.
pg4, line 151. What position was the arm in? Elbow flexed? Arm straight?, Standing? Seated?
line 155 - was the counter movement prior to the jump standardised?
line 157 - more detail required on a 4x10m shuttle test or needs referencing to the paper that details it. Otherwise, this is hard to replicate for another researcher.
line 158 - how was time measured? Stopwatch or timing gates? Was it only one person who measured?
line 159 - same again, what is the protocol for the back saver test? Needs reference or more description.
line 161 -164 - This reads like only 10% of the children could be considered to have good/optimal fitness (above the 90th percentile in this sample)??? Consider revising.
Results
There is no raw information regarding the performance on the physical fitness tests. Adding this would help other researchers replicate the results and provide information regarding expected values for these tests in a child and adolescent population. This would also give readers an understanding of what is good/great etc. Consider adding this information.
Table 1 - the latter half is not reported in the methods so I am unsure as to why it is reported in the table (All the maternal educational level data)?
Discussion
I really struggled with aspects of the discussion and was not sure on the points you were trying to make in part. I think this is a writing issue rather than anything that you said is incorrect but it does make it hard to interpret how your results add to the literature and the significance of them.
pg 8, line 272, entire paragraph - this is more of an introduction paragraph. Please consider starting by linking your findings to your aims - were your aims correct and what is the significance of them?
line 278 - there are three thoughts going on in the one sentence and it is very tough to read.
line 287, entire paragraph - I am not sure what the point you are trying to make is. It seems like you are referring to the parent's ability to correctly determine physical fitness in their kids and you are using the flexibility component responses to highlight this. however, the point is lost as there is a lot of extra information provided in this paragraph that is unclear. Consider revising
line 307 - Not sure this is the correct use of the term "conviviality".
line 310 entire paragraph - the discussion of the physical activity questionnaire is rather confusing. This section rambles a bit and could be condensed. Please consider revising.
pg 9, line 320 - 323 - awkward wording (e.g. measured subjective measure). Please revise
line 331 - I find it very difficult to consider a 20m shuttle test the gold standard for assessment of cardiorespiratory fitness when we have metabolic carts. I think this is one limitation in your study, that you didn't use gold standard physical fitness tests. But, despite this you still have moderate validity etc.
line 339 - I don't know what your point is here. Further, there is some repetitive language here (e.g. "In addition" in successive sentences).
line 346 - 348 - Are you sure about this paragraph? I am not certain that this is as definitive as you have stated. You have the data, did you look at the correlations between BP and CRF in your data set? This would strengthen your argument here.
line 349 - entire paragraph. I think this needs to be added to the paragraph prior but could be reduced by about half with good editing.
weaknesses section - What about bias and blinding of participants to results. It is entirely possible the children talked with eachother, decided to run the same amount of time etc.
Author Response
Response to Reviewer 2 Comments
Broad Comments:
Point 1: What is the rationale for including 3-year old children in the study? Is there some evidence to suggest 3-year old's current physical fitness is correlated to future health outcomes? You have pointed out this association in adolescence but their is little rationale provided as to why the children (especially the very young ones) were included.
Response 1: Thank you for this question. Physical fitness has been recognized as an important marker of health (4, 7, 8). Studies suggest that increasing physical fitness in younger children can have positive health effects (Ortega et al. Endocrinol Nutr. 2013;60(8):458-69; Leppänen et al. Med Sci Sports Exerc. 2017 Oct;49(10):2078-2085) . Castro-Piñero et al. J Pediatr. 2018; S0022-3476(18)31466-5 indicate that the development of high levels of physical fitness should begin in early childhood, which justifies the importance of evaluating the physical fitness of 3-year-old children.
Point 2: How were the participants blinded to their individual results? Did the parents communicate with the children between the questionnaire completion (first time) and the physical fitness test? Similarly, were the results of the physical fitness test (the first time) made available to participants (or their parents) between the two testing dates? How did you limit the issue of biasing the second tests?
Response 2: Thank you for this question. Participants were not informed about their performance during the tests. The parents were not present and did not communicate with their children during the tests. Participants had access to their physical fitness test results at the end of the study, he results of the measures between the first and second evaluation were not made available. Participants did not know that we were evaluating the concordance between the questionnaire and the physical tests. In order to avoid biasing the second tests, they were applied by the same evaluators, following the same procedures
Point 3: Similar to the comment above, were the physical fitness tests randomized or done in a set order each time?
Response 3: Thank you for this question. Page 4, lines: 153 – 179. All physical tests were performed individually, in the same order, as follows: handgrip test; standing broad jump test; sit and reach test; 4 x 10 m shuttle; 20 m shuttle run test.
Point 4: Did the students complete the shuttle test in groups? If so, how many were in a group, were they randomized into groups and where they were along the line to prevent running with a friend etc.?
Response 4: Thank you for this question. No. All the tests were performed individually. (Page 4, lines 153-154) All physical tests were performed individually, in the same order, as follows: handgrip test; standing broad jump test; sit and reach test; 4 x 10 m shuttle; 20 m shuttle run test.
Point 5: It is unclear how you determined physical fitness scores. You listed it as the average of the four physical fitness tests - is this the average of the percentile they were in for each test or the average of the raw score on each test? Why use the average of four - is this a fair representation of physical fitness? As there is no reference provided so I assume this is unique to your study, I think you need to justify your rationale for this a little more.
Response 5: Thank you for this question. (Page 4; lines 180-181) We use the same methodology proposed by Ortega et al. 2011. We apologize but this reference is now included in the new version of the manuscript.
Point 6: I am unsure as to why you referenced to blood pressure as not everyone with good cardiorespiratory fitness has good blood pressure. There might be a good reason for it, but it wasn't clear to me why you did this. I know you have referenced it in the discussion but the rationale should be provided prior to the discussion in my opinion so the reader understands why you have used this method.
Response 6: Thank you for your comment. We include the following sentence on the Materials and Methods section (page 3: lines 102-106) … The construct validity in this study was observed by the respondent of the physical fitness questionnaire on a health indicator(20). The health indicator chosen was high blood pressure. Because low levels of physical fitness are associated with the development of cardiovascular diseases(21). And systemic arterial hypertension is an independent risk factor for cardiovascular diseases(20).
Point 7: The tests you used for physical fitness are not gold standard (e.g. 20m shuttle run is not as accurate as a caridopulmonary exercise test for the determination of cardiorespiratory fitness). Some thought should be given to highlighting this as a limitation. Despite this, you still report good reliability and moderate validity.
Response 7: Thank you for your comment. (Page 10, lines 379-381) We include this limitation in the manuscript. Another possible limitation of the present study was to evaluate the physical fitness by field tests. However, Ortega et al (12), used these widely applied tests for validity and reliability analyses.
Specific Comments
Point 1: Abstract
Nothing to comment on. Well written.
Response 1: Thank you
Introduction:
General comments:
Point 1: Consider defining physical fitness better - specifically highlighting the parameters you measured and their association with health. I think this would strengthen the rationale.
Response 1: Thank you for your comments. We include the following sentence in the introduction (page 1, lines 40-42) “Physical fitness integrates measures of most body functions, such as skeletomuscular and cardiorespiratory, involved in the performance of daily physical activity and/or physical exercise(1). This is the reason why physical fitness is considered an important health marker(2).
Point 2: Consider adding more justification for choosing the age ranges you did, especially the young children. I don't think enough evidence has been provided to justify assessing physical fitness in 3-year olds. Just a line or two would be useful.
Response 2: Thank you for your comments. We include the following sentence on the Introduction: “……(page 2, lines 76-77) Examining physical fitness since early ages may contribute to the development of strategies to prevent the reduction of physical fitness throughout life(16).”
Point 3: p2, line 48 - What ages do you define childhood and adolescence?
Response 3: Thank you for this question.. We considered children between 3 to 10 years, and adolescents between 11 to 17 years old (17).
Point 4: line 49 - This paragraph seems to be unfinished. What is the point you are trying to make?
Response 4: Thank you for your comment. We include the following sentence on the Introduction: (Page: 2, Line 52-53) These evidences indicate that physical fitness screening is necessary in health-related educational follow-ups and in epidemiological studies(11).
Point 5: line 51 - what does a physical fitness assessment involve?
Response 5: Thank you for this question. Health-related tests involve the evaluation of cardiorespiratory endurance, muscular strength and endurance, flexibility, and body composition(2).
Point 6: line 62 - these references, 12-14, would seem to be good comparisons for your results. Have you considered comparing your results against them in the discussion (especially the children and adolescent ones)?
Response 6: Thank you for the suggestion. Our results were compared with these references. (page 9, lines 315-318 and lines 340 – 341).
Point 7: Line 66 - why 3-7 year olds? I thought you examined 3-10year olds? Maybe a typo?
Response 7: Thank you for this question. We appologize for this typo, and we corrected it in the new version of the manuscript. Sorry I did not refer to the age considered in our study. We described that the IFIS was not validated for children under 9 years of age.
Point 8: line 70 - needs referencing (other South American samples) or is this a general use of it in the public?
Response 8: Thank you for this question. We apologize for forgetting to include this reference in the manuscript. (Page 2, line 74) We included this reference in the new version of the manuscript.
Materials and Methods
Point 1: pg3, line 118, What scales and stadiometer were used? Brand name, company etc.?
Response 1: Thank you for this question. We include this information in the new version of the manuscript… (page. 3, lines 128-130)We used the ultra Slim W801 digital grams scale (Crivitta Diagnostica Ltda - São Paulo-Brazil and Wall stadiometer millimeters scale (Cardiomed - Paraná-Brazil).
Point 2: line 131 - I assume all BP was determined by averaging the two measurements, not just high BP? Consider rewording this section (you could remove the word high in this sentence and just define the acronym HBP in the following sentence.
Response 2: Thank you for the suggestion. We have modified this sentence on the Materials and Methods section. (page 3, lines 137-139; page 4, lines 140-147) We measured systolic (SBP) and diastolic (DBP) blood pressure with a calibrated mercury column sphygmomanometer coupled to an Omron device (Omron Health Care, Japan), model HEM-7200, with an appropriate cuff for each arm size, we used three different cuffs according to the following anthropometric measurements of the arm: 12 to 21 cm (small), 22 to 32 cm (medium), and 33 to 42 cm (large)(28). The measurements were performed on the right arm of the participants because of coarctation of the aorta, and the arm was supported at the level of the heart; all were placed in a quiet room. The children and adolescents were sitting, their backs resting on a chair, a relaxed arm resting on a rigid surface, and uncrossed feet resting on the floor. After 5 minutes of rest, the measurement was initiated. BP was determined by averaging two measurements taken within 5 minutes (min) of each other, with the subject resting for at least 5 min before the first measurement(29). High blood pressure (HBP) was defined as SBP and DBP above the 95th percentile for sex, age and height following the American Academy of Pediatrics protocol(30).
Point 3: pg4, line 151. What position was the arm in? Elbow flexed? Arm straight?, Standing? Seated?
Response 3: Thank you for this question. We include this information in the new version of the manuscript (page 4, lines 154-160)… (1) handgrip test (maximum handgrip strength assessment) using a hand dynamometer with adjustable grip (Jamar® PC5030J1, Fit Systems Inc, Calgary, Canada). The participant stays in a standard bipedal position with the arms in complete extension holding the dynamometer without touching any part of the body with it. Scores were calculated as the average of the right and left handgrip strength used in the analysis(26). Two trials were allowed for each hand, and the average score was recorded in kilograms (kg).
Point 4: line 155 - was the counter movement prior to the jump standardized?
Response 4: Thank you for this question. Yes. (page 4, lines 160 – 164) The standing broad jump test (lower limb explosive strength assessment) was prior to the to jump. The participant jumps as far as possible off the stand, trying to land with both feet together and maintaining the equilibrium once landed (it was not allowed to put the hands on the floor). The score was obtained by measuring the distance between the last heel-mark and the take-off line. The best of two attempts was recorded in centimeters(32).
Point 5: line 157 - more detail required on a 4x10m shuttle test or needs referencing to the paper that details it. Otherwise, this is hard to replicate for another researcher.
Response 5: Thank you for your comments. We included this reference in the new version of the manuscript. (page 4, lines 166-171) Speed-agility was assessed with the 4 x 10 m shuttle run test: test in which the participant runs as fast as possible from the starting line to the other line and returns to the starting line (10 m apart), crossing each line with both feet every time. This is performed twice, covering a distance of 40 m (4 m × 10 m). Every time the child crosses any of the lines, he/she picks up (the first time) or exchanges a sponge, which has been previously placed behind the lines. The time taken to complete the test was recorded to the nearest tenth of a second(32).
Point 6: line 158 - how was time measured? Stopwatch or timing gates? Was it only one person who measured?
Response 6: Thank you for this question. Time was measured using a stopwatch. Two researches recorded the time. (page 4, lines 149-152) To ensure the objectivity of measurements of physical fitness, every researcher involved in the fieldwork read a detailed manual of operations on the field-based fitness tests before the data collection started. In addition, all researchers participated on a training of one week to standardize and harmonize measurements of the physical fitness tests.
Point 7: line 159 - same again, what is the protocol for the back saver test? Needs reference or more description.
Response 7: Thank you for your comments. We apologize, and included this reference in the new version of the manuscript (page 4, lines 166).
Point 8: line 161 -164 - This reads like only 10% of the children could be considered to have good/optimal fitness (above the 90th percentile in this sample)??? Consider revising.
Response 8: Thank you for this question. We apologize, but this is the cut-off point for rating children (3 to 10 years old) according to their level of physical fitness.
Results
Point 1: There is no raw information regarding the performance on the physical fitness tests. Adding this would help other researchers replicate the results and provide information regarding expected values for these tests in a child and adolescent population. This would also give readers an understanding of what is good/great etc. Consider adding this information.
Response 1: Thank you for your comments. We added the results of the children. (page 7, lines 270-271) The means of performance by physical fitness tests in children and adolescents are shown in Supplementary Data 1 and 2 (page 10 and 11). The results of adolescents used the cut points pre-established by literature (26, 31, 32, 33).
Point 2: Table 1 - the latter half is not reported in the methods so I am unsure as to why it is reported in the table (All the maternal educational level data)?
Response 2: Thank you for this question. We include the following sentence on the Materials and Methods section: (page 3, lines 106 – 108) Maternal education was classified according to years of school, reported in the questionnaire. The options for maternal education according to years of school were<4 years, 4 to 8 years, 9 to 12 years, or>12 years (23).
Discussion
Point 1: I really struggled with aspects of the discussion and was not sure on the points you were trying to make in part. I think this is a writing issue rather than anything that you said is incorrect but it does make it hard to interpret how your results add to the literature and the significance of them.
Response 1: Thank you for your comments. We have modified the Discussion section. Some paragraphs were changed:
(page 8, lines 286 – 291): ”In children, the questionnaire showed moderate construct validity in the components of general conditioning, cardiorespiratory capacity, muscular strength and speed/agility. In adolescents, the questionnaire demonstrated moderate construct validity in the components of cardiorespiratory capacity, muscular strength and speed/agility. The questionnaire proved to be a sensitive method to measure physical fitness in the pediatric population.”
(page 9, lines 300 – 304); “These results show that the parents / guardians of the children understood the questions in the questionnaire, being able to respond to them in a way very close to the real one, since they are in agreement with the literature (38, 39). Studies indicate that the younger the age, the greater the flexibility (38). Flexibility tends to decline after age 17, partly as a result of a decline in physical activity and normal aging (40).”
(page 9, lines 319 – 320) “In this sense, by observing them alone and in a group, parents tend to have greater knowledge also of children who study with their children.”; (page 9, lines 323 – 328) “Therefore, many questions are needed to investigate the construct of interest, which can lead to errors, overlapping responses, reduced attention and motivation (47). On the other hand, in IFIS-LP are only five questions to construct five variables (12). Due to the nature of the very specific IFIS-LP issues, the answers are less likely to overlap or confuse, reducing memory bias and increasing the chance of bias of desired behavior.”
(page 9, lines 336 – 342) ” The findings of the present study indicate that the IFIS-LP questionnaire has an acceptable validity to measure physical fitness in children and adolescents, when compared to physical tests (criterion validity). The performance of the instrument in relation to the physical test can be explained in the literature by the correlation between the evaluated outcomes (32, 53). Recent studies have also indicated acceptable validity of the physical fitness questionnaire compared to physical tests(12, 14, 44). Our results indicate that there is a concordance of the criteria measured in the physical fitness questionnaire compared to the physical tests.”
(page 10, lines 350 – 353) “The involvement of adolescents in physical activities transforms the prognosis of this health indicator(56). Because the evolution of this health indicator can be perceived during activities of daily living (3). Our results indicating that the questionnaire can be used for the purpose of a screening test.”
(page 10, lines 358 – 362)” The questionnaire presented high sensitivity, showing good results in assessing people with good/great physical fitness even with a low prevalence of high blood pressure. Cardiorespiratory fitness is a predictor of high blood pressure [57]. Low levels of speed/agility, balance, muscle strength and cardiorespiratory fitness are significantly related to the onset of high blood pressure(59).”
(page 10, lines 379 – 381)” However, the strong of the present study was to evaluate the physical fitness by field tests. But, using the same physical tests used by Ortega et al, these tests were widely used and tested for validity and reliability(12).”
Point 2: page 8, line 272, entire paragraph - this is more of an introduction paragraph. Please consider starting by linking your findings to your aims - were your aims correct and what is the significance of them?
Response 2: Thank you for this question. This paragraph was withdrawn.
Point 3: line 278 - there are three thoughts going on in the one sentence and it is very tough to read.
Response 3: Thank you for your comments. (Page 8, lines 286 – 291) We have modified the Discussion section: “In children, the questionnaire showed moderate construct validity in the components of general conditioning, cardiorespiratory capacity, muscular strength and speed/agility. In adolescents, the questionnaire demonstrated moderate construct validity in the components of cardiorespiratory capacity, muscular strength and speed/agility. The questionnaire showed to be a sensitive method to measure physical fitness in the pediatric population”.
Point 4: line 287, entire paragraph - I am not sure what the point you are trying to make is. It seems like you are referring to the parent's ability to correctly determine physical fitness in their kids and you are using the flexibility component responses to highlight this. however, the point is lost as there is a lot of extra information provided in this paragraph that is unclear. Consider revising
Response 4: Thank you for your comments. The following phrase has been included on Discussion section (page 9, lines 301 – 305) In the group of children, flexibility was classified as one of the physical fitness components with the best performance. These results show that the parents/guardians of the children understood the questions in the questionnaire, and were able to respond to them similarly to the results of the physical test, and in agreement with the literature (38, 39). Studies indicate that the younger the age, the greater the flexibility (38). Flexibility tends to decline after age 17, partly as a result of a decline in physical activity and normal aging (40).
Point 5: line 307 - Not sure this is the correct use of the term "conviviality".
Response 5: Thank you for your comment. The following phrase has been included in the Discussion section (page 9, lines 320 – 321) In this sense, observing their children alone and in a group, parents tend to know also about the physical fitness of their child’s peers.
Point 6: line 310 entire paragraph - the discussion of the physical activity questionnaire is rather confusing. This section rambles a bit and could be condensed. Please consider revising.
Response 6: Thank you for your comment. The paragraph has been revised. The following phrase was included on Discussion (page 9, lines 323– 328) …”Therefore, many questions are needed to investigate the construct of interest, which can lead to errors, overlapping responses, reduced attention and motivation (47). On the other hand, in IFIS-LP are only five questions to construct five variables (12). Due to the nature of the very specific IFIS-LP issues, the answers are less likely to overlap or confuse, reducing memory bias and increasing the chance of bias of desired behavior.”
Point 7: pg 9, line 320 - 323 - awkward wording (e.g. measured subjective measure). Please revise
Response 7: Thank you for your comments. This paragraph was revised and the following phrase was included on Discussion (page 9, lines 335 – 341) “The findings of the present study indicate that the IFIS-LP questionnaire has an acceptable validity to measure physical fitness in children and adolescents when compared to physical tests (criterion validity). The performance of the instrument in relation to the physical tests can be explained in the literature by the correlation between the evaluated outcomes (32,53). Recent studies have also indicated acceptable validity of the physical fitness questionnaire compared to physical tests (12,14, 44). Our results indicate that there is a concordance of the criteria measured in the physical fitness questionnaire compared to the physical tests”
Point 8: line 331 - I find it very difficult to consider a 20m shuttle test the gold standard for assessment of cardiorespiratory fitness when we have metabolic carts. I think this is one limitation in your study, that you didn't use gold standard physical fitness tests. But, despite this you still have moderate validity etc.
Response 8: We appreciate your concern. With this sentence we are referring that the physical tests, for example, those that were used in this study and Ortega et al. can be used for validation of questionnaires because there is an agreement of the criteria measured in the questionnaire with the physical fitness measured by physical tests. The paragraph has been revised. (page 9, line 340-341) “Our results indicate that there is a concordance of the criteria measured in the physical fitness questionnaire compared to the physical tests.” And (page 10, line 379-381) “However, the strong of the present study was to evaluate the physical fitness by field tests. But, using the same physical tests used by Ortega et al, these tests were widely used and tested for validity and reliability(12).”
Point 9: line 339 - I don't know what your point is here. Further, there is some repetitive language here (e.g. "In addition" in successive sentences).
Response 9: Thank you for your comment. This paragraph was revised. (page 10, lines 350-353) The involvement of adolescents in physical activities can transform the prognosis of this health indicator (56), because the evolution of physical fitness can be perceived during activities of daily living (3). Our results indicates that the questionnaire can be used as a screening test.
Point 10: line 346 - 348 - Are you sure about this paragraph? I am not certain that this is as definitive as you have stated. You have the data, did you look at the correlations between BP and CRF in your data set? This would strengthen your argument here.
Response 10: Thank you for this question. (Table 1, page 6, lines 241)This statement was based on the prevalence of high blood pressure found in our study.
Point 11: line 349 - entire paragraph. I think this needs to be added to the paragraph prior but could be reduced by about half with good editing.
Response 11: Thank you for your comments. The paragraph has been revised. (page 10, lines 358 - 362) The questionnaire presented high sensitivity, showing good results in assessing people with good/great physical fitness even with a low prevalence of high blood pressure. Cardiorespiratory fitness is a predictor of high blood pressure (57). Low levels of speed/agility, balance, muscle strength and cardiorespiratory fitness are significantly related to the onset of high blood pressure(59).
Point 12: weaknesses section - What about bias and blinding of participants to results. It is entirely possible the children talked with each other, decided to run the same amount of time etc.
Response 12: Thank you for your comments. The tests were performed individually (page 4, lines 153-154).
Sincerely,
Regina Célia Vilanova-Campelo, on behalf of coauthors
PhD Candidate at Faculdade de Medicina, Universidade de Sao Paulo
